# T Cells in Colorectal Cancer: Unravelling the Function of Different T Cell Subsets in the Tumor Microenvironment

**DOI:** 10.3390/ijms241411673

**Published:** 2023-07-19

**Authors:** Ziwen Zheng, Thomas Wieder, Bernhard Mauerer, Luisa Schäfer, Rebecca Kesselring, Heidi Braumüller

**Affiliations:** 1Department of General and Visceral Surgery, Medical Center, Faculty of Medicine, University of Freiburg, 79106 Freiburg, Germany; ziwen.zheng@uniklinik-freiburg.de (Z.Z.); bernhard.mauerer@uniklinik-freiburg.de (B.M.); luisa.schaefer@uniklinik-freiburg.de (L.S.); rebecca.kesselring@uniklinik-freiburg.de (R.K.); 2Department of Vegetative and Clinical Physiology, Institute of Physiology, Eberhard Karls University Tübingen, 72074 Tübingen, Germany; thomas.wieder@uni-tuebingen.de; 3German Cancer Consortium (DKTK) Partner Site Freiburg, 79106 Freiburg, Germany; 4German Cancer Research Center (DKFZ), 69120 Heidelberg, Germany

**Keywords:** colorectal cancer, immunoscore, immune checkpoint blockade, tumor-infiltrating T cells, T cell therapy, γδ T cells, αβ T cells, NKT cells

## Abstract

Therapeutic options for metastatic colorectal cancer (mCRC) are very limited, and the prognosis using combination therapy with a chemotherapeutic drug and a targeted agent, e.g., epidermal growth factor receptor or tyrosine kinase, remains poor. Therefore, mCRC is associated with a poor median overall survival (mOS) of only 25–30 months. Current immunotherapies with checkpoint inhibitor blockade (ICB) have led to a substantial change in the treatment of several cancers, such as melanoma and non-small cell lung cancer. In CRC, ICB has only limited effects, except in patients with microsatellite instability-high (MSI-H) or mismatch repair-deficient (dMMR) tumors, which comprise about 15% of sporadic CRC patients and about 4% of patients with metastatic CRC. The vast majority of sporadic CRCs are microsatellite-stable (MSS) tumors with low levels of infiltrating immune cells, in which immunotherapy has no clinical benefit so far. Immunotherapy with checkpoint inhibitors requires the presence of infiltrating T cells into the tumor microenvironment (TME). This makes T cells the most important effector cells in the TME, as evidenced by the establishment of the immunoscore—a method to estimate the prognosis of CRC patients. The microenvironment of a tumor contains several types of T cells that are anti-tumorigenic, such as CD8^+^ T cells or pro-tumorigenic, such as regulatory T cells (Tregs) or T helper 17 (Th17) cells. However, even CD8^+^ T cells show marked heterogeneity, e.g., they can become exhausted, enter a state of hyporesponsiveness or become dysfunctional and express high levels of checkpoint molecules, the targets for ICB. To kill cancer cells, CD8^+^ T cells need the recognition of the MHC class I, which is often downregulated on colorectal cancer cells. In this case, a population of unconventional T cells with a γδ T cell receptor can overcome the limitations of the conventional CD8^+^ T cells with an αβT cell receptor. γδ T cells recognize antigens in an MHC-independent manner, thus acting as a bridge between innate and adaptive immunity. Here, we discuss the effects of different T cell subsets in colorectal cancer with a special emphasis on γδ T cells and the possibility of using them in CAR-T cell therapy. We explain T cell exclusion in microsatellite-stable colorectal cancer and the possibilities to overcome this exclusion to enable immunotherapy even in these “cold” tumors.

## 1. Introduction

### 1.1. Immune Cell Infiltration into the Consensus Molecular CRC Subtypes

The immune system is the body’s defense machinery against harmful influences originating either from an infection or from malignant transformation of the body’s own cells. The vertebrate immune system consists of a variety of innate and adaptive effector cells, e.g., granulocytes, macrophages, dendritic cells, T and B lymphocytes, etc., and specialized molecules, e.g., T cell receptors, specific antibodies, Toll-like receptors, interleukins, etc. (for an overview in immunology and cancer immunology, the interested reader is referred to [1,2,3]). In the current review, we focus on the role of T cells in the context of CRC.

CRCs harbor several mutations and epigenetic alterations and are, therefore, a very heterogeneous group of diseases. Most CRC cases are spontaneous and not inherited or familial, with specific mutations along the way from adenomas to carcinomas [4].

About 15% of all sporadic CRCs in stage II have a dysfunctional DNA mismatch repair (dMMR), leading to microsatellite instability (MSI) during DNA replication. The MMR complex identifies base-pair mismatches or insertion–deletion loops caused by DNA damage or inaccurate DNA polymerase transcription and repairs them. The MSI status can be further divided into MSI high (MSI-H) and MSI low (MSI-L) [5]. However, dMMR/MSI-H tumors harbor a vast infiltration of CD8^+^ and CD4^+^ T cells as well as macrophages and are characterized by the presence of type I interferons in the tumor microenvironment (TME) [6]. The proportion of dMMR/MSI-H tumors decreases from 15% of stage II to about 8% of stage III CRCs [7].

About 85% of sporadic CRC have no dysfunctional MMR and are classified as proficient in MMR (pMMR) and microsatellite stable (MSS).

As the infiltration of CD3^+^ and CD8^+^ cytotoxic T cells in the tumor center or the tumor margin is the best prognostic factor both in terms of relapse and in terms of overall survival in CRC, a consortium analyzed primary CRCs and established a molecular classification that also includes information about the tumor microenvironment (TME). Four consensus molecular subtypes (CMS) were established based on bulk transcriptomic sequencing [8] (Figure 1). The four subtypes are:(I)Consensus Molecular Subtype (CMS) 1 is characterized by MSI and a hypermutated profile of mismatch repair genes and BRAF genes. Most CRCs with microsatellite instability are in the CMS1 subtype [9]. Due to microsatellite instability, CMS1 tumors have a high amount of neo-antigens that are not expressed in healthy tissues. CMS1 patients have an immune infiltrate consisting of T helper 1 (T_h_1) cells, natural killer (NK) cells, CD8^+^ cytotoxic T lymphocytes (CTL) and dendritic cells (DCs). About 14% of CRCs belong to the CMS1 subtype.(II)The CMS2 subtype includes CRCs with higher chromosomal instability (CIN) and microsatellite-stable tumors. CMS2 tumors lack DCs in the tumor microenvironment, indicating that the CMS2 subtype is only poorly immunogenic, with few tumor-infiltrating immune cells. CMS2 patients are characterized by the infiltration of a few naïve CD4^+^ T cells and resting NK cells. The WNT signaling pathway correlates with T cell exclusion in CRCs [10]. Approximately 37% of CRCs belong to CMS2.(III)The CMS3 subtype is the metabolic subtype, characterized by chromosomal instability and a higher level of KRAS mutations compared with other CMS phenotypes. Like CMS2, CMS3 shows only poor immune infiltration and no immune regulatory cytokines. Tumor-infiltrating lymphocytes (TILs) consist of naïve T cells and T helper 17 (Th17) cells. About 13% belong to the CMS3 subtype.(IV)The CMS4 subtype shows a high expression of genes specific for immunosuppressive TGF-β signaling and of genes specific to regulatory T cells (Tregs) and myeloid-derived suppressor cells (MDSCs) [11]. CMS4 is also characterized by an upregulated expression of genes that encode chemokines that attract myeloid cells and T cells that produce interleukin 17 (IL-17), which are known to enhance carcinogenesis [12]. The CMS4 phenotype shows high levels of infiltrating macrophages and stromal cells. Approximately 23% of CRC tumors fall into CMS4. Although CMS4 patients show high levels of leukocyte infiltration, patients with CMS4 tumors have the worst prognosis of the four subtypes.

**Figure 1 ijms-24-11673-f001:**
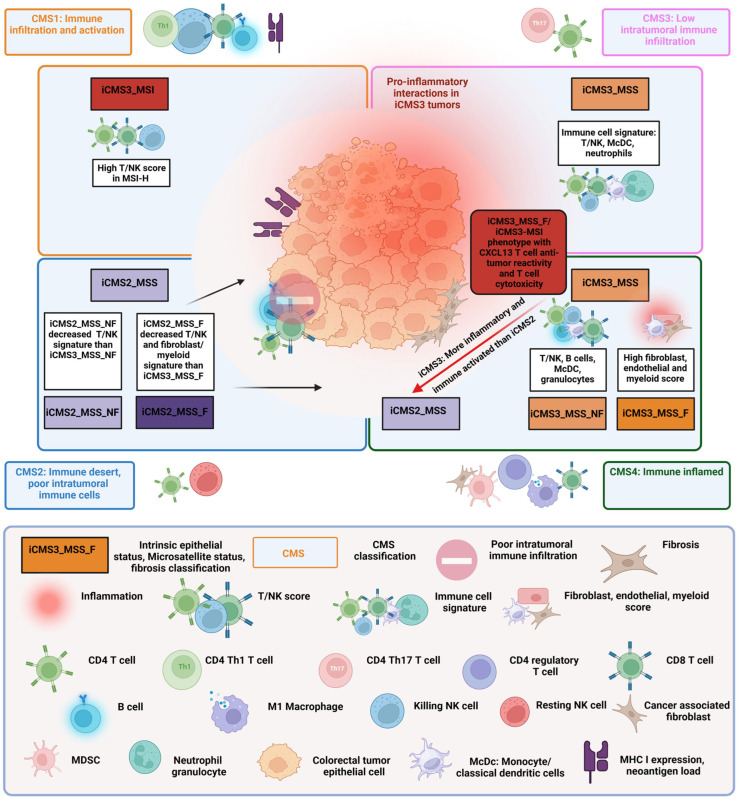
Combined transcriptome-based classification according to Guinney et al. [8] and Joanito et al. [13]. The four outer rectangles show the consensus molecular subtype (CMS) classification based on bulk transcriptomics with related immune features first described by Guinney et al. [8]. Joanito et al. refined the CMS classification with single-cell transcriptomics and described five subtypes based on the new criteria “intrinsic epithelial subtype” (I), “microsatellite status” (M) and “fibrosis” (presence of fibrosis (F) or rather nonfibrosis (NF)) (IMF). The relationship of the five new IMF subtypes i3_MSI, i3_MSS_F, i3_MSS_NF, i2_MSS_F and i2_MSS_NF to the bulk CMS subtypes is depicted in the four inner rectangles. In addition, Joanito et al. [13] show a similar i3_MSI/i3_MSS_F subtype with a significant anti-tumor cytotoxicity phenotype. For more details, see text.

The remaining 13% have mixed molecular features and do not locate in one of the four subtypes.

A recent study using single-cell sequencing identified two distinct epithelial tumor states and refined the “classical” CMS classification. Based on epithelial cells, microsatellite status and fibrosis, five functional subtypes were defined. The new intrinsic (i) CMS3 contained the “old” MSI-H CMS1 subtype and MSS tumors with epithelial features that were more similar to MSI-H cancers than to the classical CMS2 subtype. The fibrotic “old” CMS4 group comprised two epithelial subtypes, a fibrotic iCMS3_MSS_F subtype and a fibrotic iCMS2_MSS_F subtype [13]. Figure 1 summarizes the classical CMS classification and the new classification.

The classical consensus subtype classification, as well as the new “intrinsic” classification, lack accurate biomarkers of clinical outcome. A new study by Roelands et al. [14] analyzed fresh–frozen tumor samples from 348 patients with CRC using a Th1 and CD8^+^ T cell gene signature termed the immunologic constant of rejection (ICR) [15]. The ICR contains a 20-gene panel that reflects the activation of Th1 cells, the expression of chemokine ligands, activated cytotoxic signals and immunoregulatory genes. Based on the ICR data, three clusters (immune subtypes) were identified: ICR high (hot tumors), ICR medium and ICR low (cold tumors). The comparison with CMS subtypes revealed that CMS1 samples were ICR high, whereas CMS2 and CMS3 subtypes were either ICR low or ICR medium. CMS4 samples showed a high heterogeneity with a spread across all three ICR immune subtypes [14]. Analysis of the microbiome signature and combination with the ICR score identified a group of patients with superior prognosis.

### 1.2. Differences between Cold, Altered-Excluded, Altered-Immunosuppressed and Hot CRCs

Immunotherapy with checkpoint inhibition has quite remarkable clinical effects in several cancer types with a high T cell infiltration (e.g., melanoma and non-small cell lung cancer [16,17]) but show poor clinical effects in tumors that are poorly infiltrated by immune cells such as most CRCs [18]. The first description in CRC about the importance of tumor-infiltrating T cells, especially the density and location of CD8^+^ cytotoxic T cells and Th1 cells as prognostic factors, was shown in 2006 by Galon et al. [19]. Based on these findings, the concept of the immunoscore as a better prognostic factor than the pathologic tumor progression (T-stage), tumor invasion (N-stage), tumor metastasis (M-stage), TNM staging and MSI status was proposed [20]. The intra-tumor immunity can be further classified into four categories depending on the infiltration of CD8^+^ cytotoxic T cells and Th1 cells [21].

(1)The optimal immunity with a high immunoscore and hot, “T cell inflamed” tumors.(2)The altered immunity with an intermediate immunoscore and an immunosuppressed phenotype.(3)The exclusion phenotype with an intermediate immunoscore and a high density of T cells only at the margin of the tumor.(4)The phenotype with a low immunoscore and cold, “non-T cell inflamed” tumors in which other immune cells, such as myeloid cells, can be observed.

The classification of cold, altered and hot CRCs is based on the infiltration of cytotoxic T cells and T helper 1 cells and, therefore, a simplification of the complex interplay between cancer cells and innate and adaptive immune cells in the tumor microenvironment. Follicular helper T (T_Fh_) cells and B cells are also correlated with a good prognosis in CRC [22]. In most cancer entities, including CRC, the inflammatory cytokine interferon-gamma (IFN-γ), secreted by cytotoxic T cells, NK cells, γδ T cells and Th1 cells, is also associated with prolonged survival [20,23,24]. To infiltrate tumors, T cells have to be attracted by chemokines [24,25,26]. In CRCs, CD8^+^ T cells express the chemokine receptor CXCR3 that binds the chemokines CXCL-9, CXCL-10 and CXCL-11. All three chemokines recruit T cells into the tumor [27]. However, only the microsatellite instable CMS1 subtype with a high mutational load shows a high expression of T cell-recruiting chemokines [25,28].

The CMS4 subtype shows a high heterogeneity ranging from high levels of infiltrating Th1, cytotoxic immune activation and upregulated CXCR3 and CCR5 chemokine profile (“hot” tumors) to absent infiltration, no cytotoxic immune activation and no expression of chemokines (“cold” tumors) [14]. Although some CMS4 subtypes are characterized by the infiltration of T cells, these T cells are mainly not CD8^+^ cytotoxic T cells but CD4^+^ T cells that either express the forkhead box P3 (FOXP3) transcription factor [29] or produce interleukin 17 (IL-17), the so-called Th17 lymphocytes [11,30]. FOXP3-positive T helper cells function as regulatory T cells (Tregs) and interfere with effective cytotoxic immune responses. In colorectal cancer, Th17 cells are known as protumorigenic immune cells that lead to a worse prognosis [31,32]. Additionally, CMS4 tumors express chemokines such as CCL-2, CCL-5 and CXCL-12 that attract myeloid cells, leading to the infiltration of cancer-associated fibroblasts (CAFs), MDSCs and tumor-associated macrophages (TAMs) [25,33]. Thus, CMS4 tumors range from “cold” tumors with a tumor microenvironment (TME) that supports carcinogenesis and tumor growth to a “hot” TME with a favorable clinical outcome [14].

Like the CMS4 subtype, CMS2 and CMS3 subtypes show microsatellite stability and a non-hypermutated phenotype [34]. Both subtypes have an intermediate immunoscore with low T cell infiltration and immune-regulatory cytokines [28]. Compared to CMS2, CMS3 tumors show an activated WNT-β-catenin signaling pathway, correlating with T cells that are excluded from the center and accumulate at the margin of the tumor [10,35]. T cell exclusion is an important escape mechanism of “cold” tumors [36] and shows the ability of the patient to mount a tumor-specific T cell-mediated immune response and the ability of the tumor to escape the immune system by avoiding T cell infiltration [37]. CMS2 tumors are nearly devoid of T cells and other immune cells, referred to as immune deserted. This suggests an immunosuppressive TME that limits the infiltration of tumor-specific T cells and the expansion of the T cells [20].

In summary, CRC is a very heterogeneous group of diseases with different mutations in specific oncogenes and/or tumor suppressor genes, epigenetic aberrations and a highly heterogeneous tumor microenvironment. Multiple studies could show that tumor-infiltrating T cells, especially Th1 cells and CD8^+^ CTLs, are one of the most important factors both in terms of relapse and in terms of overall survival in CRC. Only 14% of all sporadic CRC patients have a highly T cell-infiltrated “hot” tumor; the majority of patients have a “cold” tumor with low or absent T cell infiltration. To obtain prognostic and clinical information, a molecular classification was introduced, the consensus molecular subtypes, consisting of four groups. The classification of patients into one of the four groups is often impossible; e.g., more than 13% of the patients harbor features of two or more groups. This led to a refinement of the classical consensus molecular subtype classification by several studies, e.g., the “intrinsic” classification or the “Immunologic Constant of Rejection” classification.

### 1.3. T Cell Subsets in CRC

Colorectal cancer is not only a disease of genetic disorders but also a complex system of non-tumor cells such as innate and adaptive immune cells and cancer cells. The tumor microenvironment (TME) is the main driver of cancer progression as well as cancer elimination. The infiltration of CD3^+^ and CD8^+^ cytotoxic T cells in the tumor center or the tumor margin, the immunoscore, is the best prognostic factor both in terms of relapse and overall survival in CRC [38]. A previous study by the International Society for Immunotherapy of Cancer (SITC) in 1885 patients with stage I–II colon cancer confirmed that even in these small tumors, the infiltration of T cells in the tumor center or the tumor margin is associated with a prolonged time of recurrence and the best overall survival [39]. The infiltration of CD3^+^ and CD8^+^ lymphocytes was a positive prognostic factor, even in patients with no detectable metastases. The same group showed that 763 patients with stage III colorectal cancer had better overall survival and a low risk of recurrence with a high immunoscore, highlighting the importance of T cells in colorectal cancer [38].

T cells are the most abundant immune cells in CRC and can be divided into CD8-positive and CD4-positive T cells. The majority of all T cells have a T cell receptor consisting of two chains, the alpha and the beta chain. After an educational phase in the thymus, where T cells learn to distinguish self from non-self and autoreactive T cells become apoptotic and die, naïve T cells circulate secondary lymphoid organs to be primed by an activated antigen-presenting cell (APC). In the tumor-draining lymph node, activated APC, e.g., dendritic cells (DCs), present tumor antigens on their major histocompatibility complex (MHC) class I to CD8^+^ T cells and on MHC class II to CD4^+^ T cells. This antigen-specific stimulation of cytotoxic CD8^+^ or CD4^+^ T cells via the T cell receptor triggers a robust immune response, leading to the generation of effector T cells [40].

#### 1.3.1. Conventional αβ CD4-Positive T Cells

The impact of conventional CD4-positive T cells with a T cell receptor consisting of an alpha chain and a beta chain in colorectal cancer is complicated, as T helper cells exist in a great variety of different subsets with both anti-tumoral and pro-tumoral functions. Additionally, CD4^+^ T cells are very plastic and can quickly shift their state due to extrinsic signals [41,42]. For example, Th17 cells can adapt to the phenotype of other T helper cells [43]. After activation by APCs, CD4^+^ T cells differentiate into effector T cells and memory T cells [44]. The activation and polarization of a naïve CD4^+^ T cell are dependent on three signals: (I) the interaction of the T cell receptor with the MHC class II complex on the APC. (II) An antigen-independent co-stimulatory signal, e.g., the CD28 molecule on the T cell with the CD80/CD86 molecules on the APC. (III) The cytokines of the environment are mainly produced by APCs.

Various co-stimulatory and co-inhibitory receptors either stimulate or inhibit cellular responses of CD4^+^ T cells. CD4^+^ T cells polarize into different subsets: the main subsets in colorectal cancer are Th1, Th2, Th17, Th22, induced or natural regulatory T cells (iTregs and nTregs) and follicular helper cells (Tfh cells). Different transcription factors (e.g., T-box expressed in T cells (T-BET) specific for Th1 lymphocytes or GATA-3 specific for Th2 lymphocytes), different cytokines (e.g., interleukin 4 (IL-4) secreted by Th2 cells or IL-17 secreted by Th17 cells), specific signal transducer and activator of transcription (STAT) proteins and different chemokine receptors define these subsets [41,45]. Table 1 and Figure 2 summarize the different subsets.

As mentioned above, T helper cells, particularly Th17 cells, can be very plastic. In inflammatory diseases, such as colitis, in vitro-generated Th17 lymphocytes can convert into IFN-γ-producing Th1 cells after adoptive transfer [54,55]. Interleukin 22 (IL-22) has been associated with resistance against chemotherapeutic drugs in patients with colorectal cancer [56] and tumorigenesis in CRC mouse models [57,58]. The cytokine transforming growth factor-beta (TGF-β) suppresses the differentiation of Th22 cells in vitro but promotes the production of IL-22 in Th17 cells in vivo [59], thus converting Th17 cells into Th22 cells. Tregs can also change their phenotype; they can lose the expression of FOXP3 and convert it into ex-Tregs, displaying Th1 or Th17 phenotypes [60].

The infiltration of Th1 cells and their derived cytokines in CRC correlates with better prognosis as Th1 cells can reduce cancer cell proliferation—partly through the induction of senescence—enhance the apoptosis of cancer cells, reduce angiogenesis and recruit cytotoxic CD8^+^ T cells [48,61,62]. Although Th1 cells activate CD8^+^ T cells and thus show an anti-tumoral effect [63], they also upregulate checkpoint inhibitors such as PD-1 on the surface of CD8^+^ T cells by the expression of IFN-γ [64].

The role of Th2 cells in CRC is controversial. Th2 cytokines, such as IL-4, IL-5 and IL-13, are pro-inflammatory cytokines that lead to chronic inflammation and inflammation-induced carcinogenesis [24]. However, Th2 cytokines can recruit eosinophils with anti-tumoral activities [65] and halt cancer progression as a result of remodeling the vasculature of the tumor [66].

Th17 cells have been found in multiple cancer types, such as melanoma, ovarian cancer and colorectal cancer, where they induce inflammatory responses [67]. Th17 cells produce IL-17, a family of cytokines with 6 subtypes, IL-17A, IL-17B, IL-17C, IL-17D, IL-17E and IL-17F, that perform distinct activities [68]. In CRC, IL-17A was shown to be involved in tumor progression and angiogenesis [68] by inducing IL-6 through the STAT3 pathway [69]. However, in contrast to IL-17A, which has clear pro-tumoral effects in CRCs, IL-17F showed anti-tumoral effects [70]. This is probably the reason why tumor-infiltrating Th17 cells are associated with a good prognosis in oral squamous cell cancer [71], gastric cancer [72] and cervical cancer [73] but not in colorectal [24,67] or hepatocellular cancer [68].

The cytokines tumor necrosis factor alpha (TNF-α) and IL-6 polarize naïve CD4^+^ T cells into IL-22-producing Th22 cells [74]. In homeostasis, IL-22 plays a prominent role in wound healing and tissue repair by inducing epithelial cell proliferation [75]. In human colorectal cancer, the infiltration of Th22 cells is associated with a good prognosis [19], although tumor-promoting Th17 cells can produce IL-22 [76].

The tumor-infiltrating lymphocytes in CRCs consist of anti-tumorigenic CD4^+^ T cells, such as Th1 or Th22 cells, and pro-tumorigenic CD4^+^ T cells, such as Th17 or immunosuppressive Tregs [77]. The balance between pro- and anti-tumoral T cells is a critical factor in colorectal cancer. The CMS4 subtype shows a high expression of genes that are specific for immunosuppressive Tregs, Th17 and MDSCs [11], and subsequently, CMS4 has the worst prognosis of the four subtypes. Treg cells express the transcription factor FOXP3 [78,79] and the IL-2 receptor alpha chain (CD25) [80]. However, the expression of CD25 is not exclusive to Tregs, as other T cells express CD25 at high levels after activation [80,81]. Depletion or reduction of Tregs leads to the upregulation of anti-tumorigenic immune responses [82]. In human colorectal cancer patients, the role of Tregs is not that clear. Some studies have shown that high numbers of Tregs are associated with a short overall survival time [83,84], while other studies showed improved survival rates [83,85]. This discrepancy could be due to the difficulties in identifying Tregs at the tumor site [77]. Tregs use several mechanisms to suppress immune responses, such as the production of inhibitory cytokines, e.g., TGF-β, IL-10 or IL-35 [86,87,88,89], or the transition of APC from a tumor-reducing to a tolerant phenotype through the checkpoint receptors cytotoxic T lymphocyte antigen 4 (CTLA-4) and lymphocyte activation gene 3 (LAG-3) [90].

To mount an effective antibody-mediated immune response in colorectal cancer, B cells need the help of follicular helper T (Tfh) cells in tumor-infiltrating lymph nodes and tertiary lymphoid structures [91]. However, the involvement of Tfh cells in solid cancer is only insufficiently described. This is due to a great variety in the definition of Tfh cells used in different studies [92]. Most studies define Tfh cells as CD4^+^, CXCR5^+^, PD-1^+^, ICOS^+^, BCL6^+^ or IL-21^+^. CXCR5 promotes the migration of Tfh cells into germinal centers [93]. BCL6 is the main transcription regulator in Tfh development [93]. Increased numbers of Tfh cells in colorectal cancer are associated with a good prognosis [94]. Tfh cells are the main producers of the cytokine Il-21 that promotes B cell activation and antibody class switching to an anti-tumor IgG1 and IgG3 class [95]. In a study using MC38 CRC cells in mice, IL-21 regulated CD8^+^ T cell responses and enhanced the secretion of IFN-γ and granzyme B [95].

In summary, T helper cells differentiate into several subsets with divergent functions in the tumor microenvironment that range from anti-tumorigenic effects mediated by Th1 cells, Th22 cells or follicular helper cells to tumor-promoting effects mediated by Th17 cells and regulatory T helper cells. In colorectal cancer, the CMS4 subtype that often shows a high infiltration of Tregs and Th17 has the worst prognosis of all four subtypes, whereas subtypes with a high infiltration of Th1 cells have a much better prognosis.

#### 1.3.2. Conventional αβ CD8-Positive T Cells

CD8^+^ CTLs are effector cells of the adaptive immune system that recognize antigens presented by MHC class I molecules on the surface of APCs, typically dendritic cells (DCs). CD8^+^ cytotoxic T lymphocytes are the main T cell subset in cancer as they directly kill malignant target cells. The killing of the target cell is quite fast as CD8^+^ T cells store cytotoxic molecules, such as granzyme, perforin or FasL, in preformed vesicles. Naïve T cells express L-selectin on their surface, which guides the naïve T cell from the blood into secondary lymphoid tissues, including tumor-draining lymph nodes [96]. In the tumor-draining lymph node, CTls, such as CD4^+^ helper T cells, need three signals from the APC to become activated and polarized into an effector cell. However, CD8^+^ T cells require more co-stimulation than CD4^+^ T cells to turn into effector cells. CD4^+^ helper cells provide this additional help. After priming in the lymph node, CTLs traffic through the blood into the tumor. CTLs in the TME of CRC produce high amounts of IL-2, IL-12 and IFN-γ that activate the killing efficiency of NK cells and CTLs and enhance the expression of the chemokines CXCL-9, CXCL-10 and CXCL-11. The corresponding chemokine receptor CXCR3 is expressed on the surface of CD8^+^ T cells. After binding the chemokines, CD8^+^ T cells infiltrate the tumor. As CD4^+^ Th1 cells also express the chemokine receptor CXCR3, Th1 cells enter the tumor and become activated. Thus, CD8^+^ T cells not only kill target cells (e.g., cancer cells) but also help change the TME into a more tumor-suppressive environment. CTLs, therefore, have a great impact on the survival of CRC patients [97]. Microsatellite instability leads to the synthesis of neo-antigens by the cancer cells [98,99]. Patients with MSI status, due to the expression of neo-antigens by colorectal cancer cells, show a much higher CTL infiltration and a better prognosis than MSS patients [100].

Tumor-infiltrating lymphocytes (TILs), especially CD8-positive T cells, express various amounts of co-inhibitory and inducible receptors, which interfere with the co-stimulatory signal needed to activate T cells. Well-known checkpoint receptors are CTLA-4, programmed death 1 (PD-1), T cell immunoglobulin and mucin-containing protein 3 (TIM-3) [101]. The upregulation of inhibitory checkpoint molecules on CTLs and the increased expression of the corresponding ligands on colorectal cancer cells as well as myeloid cells in the TME lead to the suppression in the proliferation and cytokine production of the T cell and finally to dysfunctional and exhausted T cells [102,103].

Figure 3 and Table 2 summarize the characteristics of cytotoxic CD8^+^ T cells.

**Figure 3 ijms-24-11673-f003:**
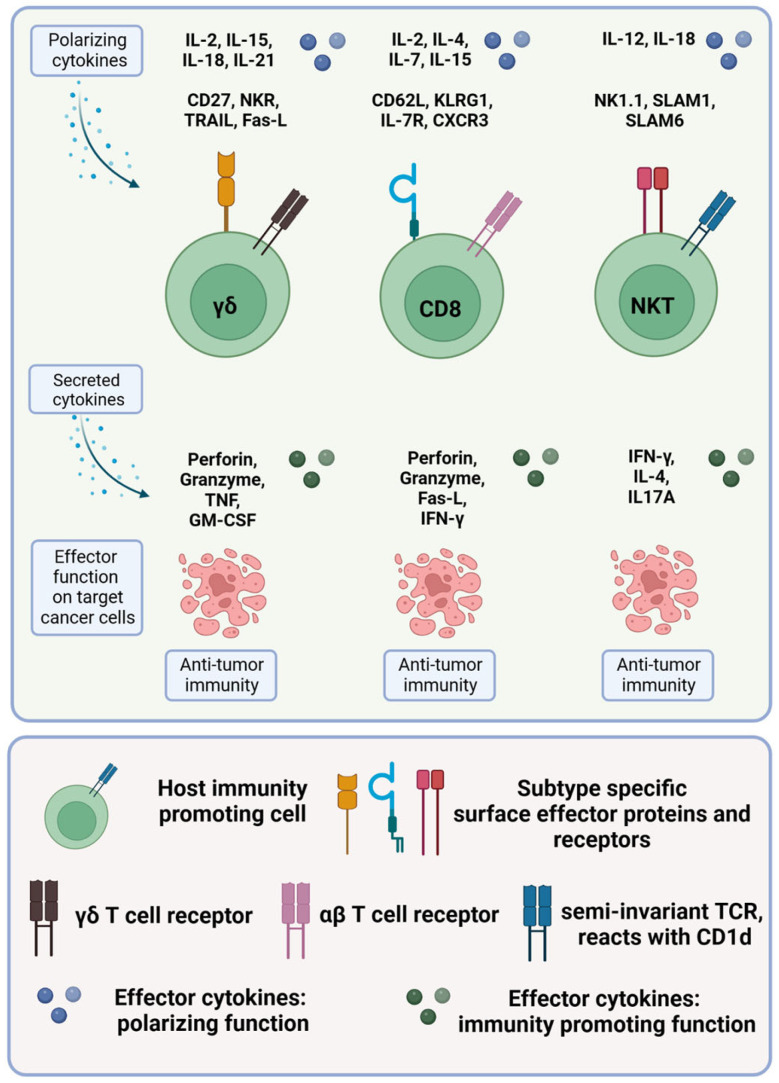
Cytotoxic T cells in colorectal cancer. The different immunity-promoting killer cell subsets in colorectal cancer are shown in green. The polarizing cytokines (first row), the morphology, including the specific surface proteins and receptors (second row), the secreted cytokines (third row) and the respective effector functions (fourth row) of the different immune cell subsets are given. For more details, see text.

#### 1.3.3. γδ T Cells

T cells with an αβ T cell receptor recognize antigens present on MHC class I or class II by APCs and are tolerant towards self-antigens. In contrast, T cells with a γδ TCR do not rely on MHC presentation and do not discriminate between self and non-self. CD8^+^ cytotoxic T cells with an αβ T cell receptor are the main targets for immune checkpoint inhibition as these immune cells recognize tumor neo-antigens presented on MHC class I. Therefore, in CRCs, checkpoint inhibition is restricted to tumors with a high mutational burden. A recent article questions this hypothesis. In MMR-deficient CRCs from 70 patients with mutations that lead to the loss of MHC I, immune checkpoint blockade activated γδ T cells and led to clinical benefits in 95% of the patients compared with 62% clinical benefit in patients with wildtype MHC I [104]. However, γδ T cells, with their different subsets and different physiologic functions, can recognize molecules expressed by stressed cells independently of peptides, lipids or metabolites [105].

Like the αβ TCR, the γδ TCR assembles with the CD3 adaptor protein of the T cell receptor. The variable regions of the γδ TCR are structurally more like the immunoglobulin domains of antibodies [106] and recognize antigens on the surface of cells rather than antigens presented on MHC complexes [107]. Surprisingly, most of the circulating γδ T cells in human blood are oligoclonal, predominantly with a variable region 9 of the γ chain (Vγ9) and a variable region 2 of the δ chain (Vδ2) [108]. Infections like those with the virus CMV lead to a massive oligoclonal expansion of individual clones [109,110,111], resulting in individual γδ T cell repertoires dependent on the history of infections [108]. Several subsets of γδ T cells are organ specific, e.g., intraepithelial γδ T cells often display a restricted usage of Vγ but diverse Vδ chains dominated by Vγ7-positive and Vγ4-positive T cells in human intestinal epithelia [112]. Vδ1 T cells are the main γδ T cells in the thymus and mucosal epithelia, where they lyse infected and transformed cells [113,114]. As γδ T cells recognize more than one antigen on the surface of a tumor cell, this has led to the concept that γδ T cells can recognize stress-induced metabolic changes. γδ T cells seem to be able to sense malignant cells with very few mutations by recognizing the different metabolic changes, making γδ T cells predisposed to early cancer immune surveillance [115].

In the gut mucosa, γδ T cells, predominantly Vγ4 and Vγ7 T cells, account for 20–30% of the intraepithelial lymphocytes (IELs) in humans and about 50% in mice [116]. Several molecules, such as CD103 or integrin β7, control the migration and localization of IELs [117,118]. Both αβ and γδ IELs express cytolytic molecules, the apoptosis-inducing Fas ligand and molecules associated with natural killer (NK) cells such as NKG2A, NKG2D, NKp46 and NK1.1 constitutively [105]. In addition to the resident IELs, other γδ T cells arrive via the blood in the lamina propria [112]. These infiltrating T cells either show a cytotoxic effector-like phenotype or express the cytokines IL-17 or IL-22.

In CRC, there are two main subsets of γδ T cells. One subset produces IFN-γ and displays anti-tumorigenic functions. In addition, the other subset of γδ T cells produces IL-17A and supports tumor growth [114]. The γδ T cells that express IFN-γ are predominantly T cells with a Vγ1 TCR, whereas IL-17A producing γδ T cells are predominantly Vγ6 positive [119].

Several studies in mice indicate that γδ T cells play a protective role in the early phases of CRC [120,121]. In human CRCs, the relevance of infiltrating γδ T cells is not so clear and seems to depend on the subtype. Tumor-infiltrating γδ T cells that express IL-17A promote cancer growth [122,123], whereas γδ T cells that express IFN-γ efficiently kill colorectal cancer cells [113,114]. Human tumors are frequently infiltrated by Vγ9Vδ2 T cells [124], and patients with high infiltration of Vγ9Vδ2 show a better survival [125]. The superior cytotoxic capacity of Vγ9Vδ2 T cells was shown by the production of higher levels of NKG2D, granzyme B, Fas ligand and several cytokines and chemokines compared with αβ T cells or NK cells [126] and the efficiency to kill a variety of CRC cell lines in vitro [124]. Interestingly, Vδ1 T cells isolated from the blood of CRC patients and expanded ex vivo show an increased cytotoxic activity compared with Vδ2 T cells [127]. In human CRCs, γδ T cells with a Vδ1 TCR are the main subset, indicating that the tumors cannot negatively influence the γδ T cell infiltration. Some tumors and virus-infected cells express the stress-induced proteins MHC class I-related chains A and B (MICA and MICB) [128]. Vδ1 T cells recognize MICA and MICB on the surface of the tumor cell and become activated. Additionally, Vδ1 T cells do not need activation via the TCR but can be activated via ligation of the stimulatory receptors such as NKG2D or NKp30 [129].

Most studies focus on the anti- or pro-tumorigenic functions of the infiltrating γδ T cells, but this is a very static approach. A new study examined the dynamic functions of γδ T cells in cancer development and progression [130]. In a preclinical mouse model of CRC, Reis et al. [130] showed that in small tumors, cytotoxic Vγ7 and Vγ1-positive cells that produce IFN-γ are essential for tumor surveillance, whereas later, when the tumor grows, IL-17-producing Vγ6^+^ T cells promotes tumor growth. The study of Reis et al. is quite important, but the translation from the CRC mouse model to the clinical situation is complicated. There are no direct counterparts of the murine γδ T cell subsets in humans [119]. However, human γδ T cells are less susceptible to expressing IL-17 than mouse γδ T cells [119]. In the same study, analysis of infiltrating γδ T cells from non-tumor areas and tumors from CRC patients showed an enriched gene signature of some genes such as CD9 and LGALS3 that were described as associated with murine IL-17-producing γδ T cells [131] but no upregulation of IL-17A in the infiltrating T cells [130].

Figure 3 and Table 2 summarize the characteristics of γδ T cells.

#### 1.3.4. NKT Cells

NKT cells have characteristics of NK cells, such as the expression of CD16 and CD56 and the production of cytolytic molecules, but they express a TCR [132]. Although NKT cells express an αβ T cell receptor, their phenotype and functional properties are very different from classical T cells. The T cell receptor of NKT cells recognizes lipid antigens on the surface of the non-classical MHC class I molecule or the CD1d molecule but not as conventional T cells that are activated by antigens presented on MHC [133]. CD1d is expressed on most nucleated cells and, like classical MHC I molecules, is often suppressed in cancer cells [134]. NKT cells divide into two subtypes, Type I NKT cells—also called invariant NKT cells (iNKT cells)—and Type II NKT cells. Type I NKT cells have a semi-invariant TCR, often Vα24Jα18, and can be further divided into five distinct functional subsets based on the expressed cytokine profile.

After stimulation, the Tfh-like NKT cell expresses IL-21; the IL-17-like NKT cell expresses IL-17, IL-21 and IL-22; the Th1-like type expresses IFN-γ and TNF-α; the Th2-like NKT cell expresses IL-4 and IL-13; and the Treg-like NKT cell expresses IL-10. The production of cytokines is very fast in response to stimuli [135]. Th1-like NKT cells have anti-tumorigenic functions, whereas Th2-like NKT cells protect against autoimmunity [136]. Th1-like NKT cells produce IFN-γ and activate effector NK cells and CD8^+^ T cells [137]. Although iNKT cells can have immunosuppressive effects, e.g., the production of IL-10 by Treg-like NKT cells, iNKT cells show predominantly anti-tumorigenic functions in a variety of tumor models [136]. The infiltration of iNKT cells in colorectal tumors of 103 patients led to a better prognosis than tumors that were infiltrated with type II NKT cells or without NKT cell infiltration [138]. A study in a murine CRC model showed that the absence of iNKT cells did not reduce tumor growth, as type II NKT cells suppress the functions of infiltrating immune cells [139,140]. A recent study in CRC patients showed that high frequencies of tumor-infiltrating PD-1-positive NKT cells led to significantly longer disease-free survival [141].

In contrast to type I NKT cells, type II NKT cells have a heterogeneous TCR repertoire and recognize a diverse range of lipid antigens. Type II NKT cells have a more immune-suppressive effect in several tumors [132,140].

Figure 3 and Table 2 summarize the characteristics of NKT cells.

**Table 2 ijms-24-11673-t002:** Characterization of the T cell receptor (TCR)-expressing immune cell subsets in colorectal cancer.

Name	Surface Marker	Polarizing Cytokines	Function	Secreted Cytokines	Reference
CTL	CD8^+^ CCR4^+^ CXCR3^+^ IFNγR^+^ αβ TCR	IL-2, IL-7, IL-4, IL-15	Destruction of tumor cells by cytotoxic molecules, e.g., perforin, granzymes, granulysin and Fas ligand.	IFN-γ, TNF-α, IL-2	[142,143]
iNKT	CXCR3^+^, NK1.1^+^ Vα24-Jα18 TCR	IL-12, IL-18	Anti-tumoral functions mainly mediated by IFN-γ, TNF-α and downstream activation of NK cells and CTLs. Destruction of tumor cells by cytotoxic molecules.	IFN-γ, TNF-α, IL-2, IL-4, IL-5, IL-3, IL-13, IL-10, IL-17, IL-21, IL-22	[132,136]
Type II NKT	αβ TCR		Pro-tumoral functions mediated by IL-4 and IL-13.	IL-4, IL-13	[132,136]
γδ T1	γδ TCR CD27^+^ CD122^+^ CD45RB^+^ Fas-L^+^	IL-2, IL-15	Anti-tumoral functions mainly mediated by IFN-γ, TNF-α and downstream activation of NK cells and CTLs. Destruction of tumor cells by cytotoxic molecules.	IFN-γ, TNF-α, IL-2	[119,144]
γδ T17	γδ TCR CCR6^+^ SCART-2^+^	IL-23, IL-1β, IL-6, TGF-β	Regulator of epithelial barrier integrity.	IL-17	[145,146]
γδ T2	γδ TCR	IL-4	Maintain tolerance to self-antigens, prevent the induction of auto-antibodies. Suppression of effector T cell-mediated immune responses in colorectal cancer.	IL-4, TGF-β	[147]

In summary, tumor-infiltrating cytotoxic CD8^+^ T cells are the main effector cells in cancer control and have a great impact on the survival of CRC patients. They recognize neo-antigens presented by MHC class I molecules on the surface of colorectal cancer cells, kill their target cells and help change the TME to a more tumor-suppressive environment. Cancer cells often downregulate MHC class I and are invisible for CTLs. CRCs that downregulate their MHC class I can be recognized and killed by γδ T cells and NKT cells that are MHC class I-independent but recognize molecules expressed by stressed cells such as cancer cells.

## 2. Tumor Initiation: Recognition of Colon Cancer Cells by T Cells

The classical model of cancer initiation envisaged a normal cell transforming into an atypical or dysplastic cell with progression into an invasive or malignant cell. Initiation occurs when genetic, metabolic and carcinogenic factors damage the DNA molecules, inducing mutations in oncogenes or tumor suppressor genes, leading to uncontrolled cell cycle progression and inactivation of apoptosis. There has to be a series of mutations prior to pathologically observable morphological changes. There are multiple bottlenecks that malignant cells must overcome to successfully form a tumor, many of which depend on subverting normalizing cues from the surrounding tissue, followed by hijacking microenvironmental processes to support the initiation of the tumor [1]. About 50 years ago, Burnet and Thomas proposed the cancer immunosurveillance hypothesis [148,149]. It stated that the immune system of the host recognizes antigens of newly arising tumors and eliminates these tumor cells before they become clinically evident [150]. Mutations within cancerous cells can be detected by immune surveillance and initiate the immune response to eliminate “non-self” cells. In most cases, these cells are cleared by the immune surveillance system [151].

Progressive cancer was seen as a rare event in which tumors evade immune surveillance [152]. However, there is increasing evidence that tumor elimination represents only one dimension of the complex relationship between the immune system and cancer. Work from several groups showed that in addition to cancer immunosurveillance, the immune system not only controls tumor quantity but also its quality (immunogenicity). Immune selection pressure favors the development of less immunogenic tumors, which escape recognition by a functioning immune system and a process termed immunoediting [153]. Cancer immunoediting is composed of three phases: elimination, equilibrium and escape.

The initiation phase of cancer has not yet been studied extensively, largely due to the absence of methodological tools for detecting occult lesions and analyzing the effects of immunity on tumor initiation. However, there are many mechanisms that lead to the early elimination of tumors. Danger signals triggered by cancerous cells may elicit immune responses. Upon DNA damage in precancerous cells, stress-induced ligands are induced, which can be sensed by the lymphocyte activation receptor, NKG2D, and the surrounding cells are alerted. Further danger signals released by transformed cells, e.g., cytokines and heat-shock proteins, may provide sufficient signals to activate the immune system [153]. Most of these danger signals are monitored by DCs, which become differentiated and activated and crosstalk with NK cells and later T cells. NK cells, γδ T cells and NKT cells may be immediately recruited to the danger site.

During the early stages of tumor initiation, if enough immunogenic antigens are produced, naïve T cells will be primed in the draining lymph nodes, followed by their concomitant activation and migration to the TME. From there, they mount a protective effector immune response, eliminating immunogenic cancer cells [154]. T cells and NKT cells may recognize developing tumor cells via TCR interaction and employ cytotoxic effector mechanisms to eliminate the transformed cells [155]. During the early stages of carcinogenesis, i.e., tumor initiation, cytotoxic immune cells—mainly NK cells and CD8^+^ T cells—recognize and eliminate arising immunogenic cancer cells [1].

Tumor cells evade the immune system by decreasing immune cell recognition, inducing immune cell dysfunction or inhibiting immune cell infiltration into the tumor. One tactic is to inactivate or downregulate the major histocompatibility complex or its associated machinery on cancerous cells. This would allow the cell to evade recognition by T cells. In addition, tumor cells may acquire resistance against the cytotoxic functions of immune cells, such as the expression of anti-apoptotic molecules, preventing tumor cell death. Furthermore, especially in colorectal cancer, it has been shown that tumors are surrounded by a complex immunosuppressive network [155]. The crosstalk between tumor cells and immune cells establishes this potent immunosuppressive milieu consisting of VEGF, TGF-β, IL-10, PGE_2_ soluble phosphotidylserines, soluble Fas or IDO. Furthermore, tumors can induce the recruitment of immunosuppressive immune cells, such as regulatory T cells and myeloid-derived suppressor cells.

## 3. Tumor Progression: When Cancer Cells Escape T Cell Recognition

### 3.1. T Cell Exclusion

In CRC, it has been shown that the exclusion of CD8^+^ T cells from cancer cells correlates with poor prognosis [19,156]. Several mechanisms are now known to achieve this local immune suppression within the TME of solid tumors, including CRC, through T cell exclusion [36]. First, T cells rely on appropriate signaling to migrate to peripheral tissues. Downregulation of chemokines involved in T cell recruitment, such as CXCL-9 and CXCL-10, as well as alterations in chemokine signaling, e.g., via nitration of CCL-2, results in reduced T cell infiltration [18,36,157]. Downregulated expression of adhesion molecules (ICAM-1, VCAM-1) on endothelial cells of the tumor vasculature prevents extravasation of T cells and, thus, infiltration [158,159,160]. Additionally, the endothelial expression of FasL, which, in turn, is induced by immunosuppressive agents produced by the TME (VEGF, PGE_2_, IL-10), also inhibits T cell extravasation [161]. Stromal barriers, which are caused, for example, by dense ECM that is, in turn, deposited by cancer-associated fibroblasts (CAFs), contribute to the failure of T cells to migrate into the tumor core, thus preventing direct contact with cancer cells [36,162]. Furthermore, loss of HLA class I expression of cancer cells leads to limited T cell infiltration due to reduced recognition. The hypoxic milieu in the tumor tissue also limits T cell infiltration. In addition, non-tumor cells in the TME can not only impair the function of T cells per se but also prevent infiltration. Overall, T cell exclusion is achieved by lack of tumor cell recognition and thus lack of T cell recruitment, alterations in chemokine signaling for T cell recruitment, as well as impaired extravasation and migration of T cells within the TME.

### 3.2. Dysfunctional T Cells

T cell dysfunction in colorectal cancer and other solid tumors refers to a condition in which T cells have limited ability to effectively recognize and eliminate cancer cells within the tumor microenvironment [142]. The limitations concern T cell activation and survival, but also effector functions such as the production of cytokines (IL-2, IFN-γ, TNF-α) and cytotoxic molecules (granzymes, perforins), as well as proliferation, are impaired [163]. A complex interplay of multiple mechanisms can cause T cell dysfunction or impairment and thus limit their anti-tumor activity. Tumor cells or tissue-resident non-tumor cells can inhibit T cell function through the secretion of immunosuppressive factors, e.g., the cytokines IL-10 and TGF-β [18,64]. Recruitment of inhibitory cells such as Tregs, tumor-associated macrophages (TAMs), MDSCs and CAFs further contributes to attenuating the immune response and disrupting T cell function [18]. They contribute to the secretion of IL-10 and TGF-β but also display other inhibitory functions. MDSCs, for example, produce nitric oxide, reactive oxygen species (ROS), arginase 1 and indoleamine 2–3 dioxygenase-1 (IDO-1), which impair proper T cell function [64]. Recently, it has been shown in CRC that tumor-infiltrating neutrophils are able to suppress T cell function via secretion of the Matrix Metalloproteinase 9 (MMP9), which, in turn, activates latent TGF-β [164]. The increased expression of inhibitory receptors, such as PD-1, CTLA-4, TIM-3 and LAG-3, causes an upregulation of immune checkpoints and, thus, a decreased T cell activation. Metabolic changes in the tumor microenvironment in the sense of nutrient deprivation and the formation of different metabolites also alter T-cell function [142,165]. Hypoxia and acidosis are characteristic features, as well as altered glucose levels and a change in amino acid composition. In addition, T cells are subject to persistent stimulation by tumor antigens, which, similar to chronic inflammation, impairs their functionality. Cancer cells, in association with their TME, can lead to an increased presence of soluble suppressive mediators, the presence of inhibitory non-tumor cells, metabolic alterations and a change at the transcriptional level of T cells, thus resulting in impairment of T cell function.

### 3.3. Tolerance, Anergy and Ignorance

During T-cell development, T-cells that recognize self-antigens with a high affinity are eliminated by central tolerance mechanisms. Furthermore, in the periphery, different mechanisms exist to eliminate or inhibit auto-reactive T cells [142]. These mechanisms are essential to prevent autoimmunity but can also be used by tumors to evade the immune response. Cancer cells either produce neo-antigens due to mutations or self-tumor antigens, for example, gene-amplified oncogenes. The latter induces an attenuated T cell response as these are self-antigens [18]. Furthermore, tumor cells are able to downregulate specific tumor-associated antigens, thus reducing recognition by T cells. Loss of HLA class 1 expression by colorectal cancer cells also impairs T cell recognition [18]. The upregulation of inhibitory receptors, such as PD-1 and CTLA-4, on T cells leads to suboptimal activation and, thus, impaired anti-tumor activity [64]. For optimal T cell activation, stimulation of the TCR is necessary, as well as the activation of a co-stimulatory receptor (e.g., CD28). If naïve T cells encounter an antigen presented by a cancer cell or an APC that lacks a co-stimulatory ligand (e.g., CD80/86), they are not ideally activated and, thus, ineffective in the detection and elimination of tumor cells. The expression of immunosuppressive factors, such as IL-10 and TGF-β, as well as continuous stimulation of T cells by tumor antigens, results in impaired T cell function and, thus, ineffective clearance of cancer cells. Altogether, the mentioned mechanisms lead to either impaired recognition of target cells, decreased activation or failed effector functions of T cells, resulting in tolerance to cancer cells.

Anergy is the state of unresponsiveness of an antigen-specific T cell after a stimulus [166]. Anergic T cells are unable to produce cytokines, the main effector molecules. Naïve T cells, both CD4^+^ and CD8^+^, need a co-stimulatory signal, e.g., the CD28 molecule, to be activated and become effector T cells. In the absence of the co-stimulatory signal, the T cell is not efficiently activated, despite the TCR engagement with the MHC complex on the APC. Physiologically, this can happen when APCs have not received the inflammatory signals necessary for the up-regulation of the co-stimulatory molecules [167,168].

Immunological ignorance refers to a state when T cells become ignorant of their cognate antigen [142]. This can happen when self-antigens are expressed at a very low level or in immune-privileged sites, leaving the T cell in a phenotypically naïve state [169]. Tumor neo-antigens in very small tumors can induce immunological ignorance as well. As a result, tumor cells are embedded within normal tissue without detection by T cells [150,170]. In “hot” tumors with a high expression of neo-antigens in the tumor and the tumor-draining lymph node, the ignorance of the T cells can be overcome [171].

### 3.4. Exhaustion

After activation of naïve CD8^+^ T cells, T cells differentiate into memory and effector cells with high cytotoxic capacity. The effector T cells undergo apoptosis after the clearance of the antigen, while the antigen-specific memory T cells still exist even in the absence of the antigen. Memory T cells that circulate through secondary lymphoid tissues, such as lymph nodes, known as central memory T cells (T_CM_), show weak cytotoxicity with a low expression of cytotoxic molecules, such as granzyme B and a strong expansion ability. In peripheral tissues or circulation, memory T cells still show a high cytotoxic capacity with high expression of cytokines and chemokine receptors. These peripheral memory T cells are known as effector memory T cells (T_EM_). In addition, a third memory T cell subset, the resident memory T cells (T_RM_), exists. T_RM_ cells are located in non-lymphoid tissues and do not enter the blood [172].

Naïve T cells are restricted to secondary lymphoid organs where they have to recognize the right antigen and become activated. After antigen-driven activation, the T cell has to upregulate chemokines and integrins that facilitate the trafficking of the activated T cell to other organs [173]. In the tumor-draining lymph node, the sustained stimulation of the T cell receptor leads to the generation of precursor-exhausted T cells (T_PEX_) that express high levels of the transcription factor T cell factor 1 (TCF1) [174]. To successfully infiltrate a tumor, the T cell has to upregulate several integrins, and the chemokine receptors CXCR3 (the receptor for the chemokines CXCL-9, CXCL-10 and CXCL-11) and CCR5 (the receptor for the chemokines CCL-3, CCL-4 and CCL-5) [175] and the cells of the tumor have to produce the corresponding chemokines and receptor ligands. In contrast to ignorance, where low amounts of antigens lead to the absence of activation of naïve T cells, exhausted T cells were stimulated by high amounts of neo-antigens over a longer period. In the early phase of T cell exhaustion, the production of cytokines and cytotoxic molecules decreases. This decrease of effector functions is mediated by inhibitory receptors that prevent over-activation of the T cell, thereby preventing immunopathology and autoimmunity. The existence of T_PEX_ cells shows that exhaustion is not a fixed state but a dynamic process [176], leading to terminally exhausted TILs.

In cancer, a sort of chronic inflammation with the presentation of tumor antigens persists, and multiple inhibitory receptors, such as CTLA-4, PD-1, PD-L1, LAG-3 and Tim-3, are permanently expressed, especially in MSI-H CRCs with a high immunoscore. Exhaustion is a dysfunctional state, first observed in chronic viral infection where CTLs first lose proliferation capacity and IL-2 expression, then the production of TNF-α and then IFN-γ production [177,178]. T cells in the tumor are permanently activated and consequently upregulate genes related to the activation of the cell cycle and co-inhibitory receptors [178]. Terminally exhausted T cells remain unresponsive to checkpoint inhibitor blockade (ICB) and show high cytotoxicity but reduced survival, while precursor-exhausted T cells retain stem-like properties and respond to ICB [179]. The transcription factor TCF-1 is essential for T cell development [180]. In terminally exhausted T cells (T_EX_) cells, TCF-1 is epigenetically silenced after three rounds of division [176]. However, due to the stem-like phenotype of T_PEX_ cells, TCF-1 is not epigenetically silenced and remains active [178]. Terminally exhausted T cells in the tumor lose the ability to proliferate and to produce cytokines and cytotoxic molecules even after checkpoint inhibitor therapy [181].

In summary, cancer cells use countless escape mechanisms to avoid immune surveillance and anti-tumor immunity. One mechanism in CRC is the induction of terminally exhausted T cells. Immunotherapy with checkpoint inhibitor antibodies can re-activate precursor-exhausted T cells. However, checkpoint inhibitor therapy is only effective in a minority of CRCs with microsatellite instability, a high mutational burden and a high infiltration of CD8^+^ T cells. The majority of CRCs either exclude T cells from the tumor or the tumor is T cell deserted or infiltrated by tumor-promoting Th17 and Treg cells. Other escape mechanisms CRC cells use are the induction of tolerance, anergy and ignorance.

## 4. Immunotherapeutic Interventions

Metastatic CRC is a malignant disease with a relatively poor prognosis, especially when peritoneal metastases are involved [182]. Historically, the median survival of CRC patients with peritoneal metastases was in the range of 3 to 6 months. In patients undergoing cytoreductive surgery in combination with intraperitoneal chemotherapy, the overall survival clearly increased to values of 16 to 51 months, depending on the different clinical settings of the respective studies (for review, see [183]). However, those still unsatisfying clinical outcomes made it necessary to look for alternative methods to fight metastatic CRC. With the development of immunotherapies, a new anticancer therapy arose after more than 50 years of basic research (for review, see [184]). Interestingly, this new treatment regimen does not directly target the cancer cells but instead aims to re-activate the immune system, namely the different T cells, to attack the neoplastic tumor cells. Here, we focus on clinical approaches to therapeutically use the immune system to eliminate malignant CRC.

### 4.1. Immunotherapies with Immune Checkpoint Inhibitors

Most immunotherapies are based on the idea that cancers escape efficient surveillance by the immune system [63,185,186]. This escape might be therapeutically counteracted either by an enhancement of the activation phase of the immune response, e.g., by transfer of anticancer T lymphocytes, or by inhibition of the termination phase of the immune response, e.g., by the use of immune checkpoint inhibitors [187]. The latter paved the way for the concept of an immune checkpoint blockade by specific monoclonal antibodies (mAb), which has become one of the main pillars of anti-melanoma and anti-advanced non-small cell lung cancer treatment (for a summary of approved checkpoint blockade therapies, see [188]). One reason for the poor immunogenicity of tumors is the missing expression of co-stimulatory molecules or signals that mediate the full activation of T cells. In this line, CTLA-4 has been described as a second counter-receptor for CD28-mediated co-stimulatory signal transduction, thereby acting as a negative regulator of T cell activation and monoclonal anti-CTLA-4 antibodies have been shown to result in the rejection of tumors [189]. Another important negative co-receptor that is expressed on antigen-stimulated T cells and B cells is PD-1. As PD-1 is directly involved in the regulation of exhausted T cells, and as PD-1-deficient mice only present a mild autoimmune phenotype, this molecule was also recognized as a powerful target for immunological therapy with a high-efficiency profile to treat cancer [190]. In the meantime, a whole bunch of monoclonal antibodies targeting different effector molecules of negative immune regulation (immune checkpoints) were developed and tested in clinical studies. To date, immune checkpoint blockade was approved or at least validated in clinical studies for the following monoclonal antibodies:(i)Ipilimumab, an antibody that binds CTLA-4, for melanoma treatment [191].(ii)Nivolumab, an antibody that binds PD-1, for melanoma treatment [192].(iii)Avelumab, an antibody against PD-L1 targeting the PD-L1/PD-1 pathway, for metastatic Merkel cell carcinoma [193].(iv)Atezolizumab, an engineered humanized immunoglobulin G1 antibody against PD-L1, for metastatic urothelial carcinoma [194].(v)Urelumab, an anti-CD137 antibody [195], for the treatment of different solid cancers, such as bladder cancer, renal cell cancer, colorectal cancer, gliosarcoma, etc.(vi)Relatlimab, an anti-lymphocyte activation gene 3 (LAG-3) antibody in combination with nivolumab, for the treatment of resectable melanoma [196].(vii)Lirilumab, an antibody that blocks the killer immunoglobulin-like receptor (KIR)/human leukocyte antigen-C (HLA-C) interaction, for the treatment of patients suffering from myeloid malignancies [197].

The efficiency of the immune checkpoint blockade in CRC patients has already been tested in the early days of immunotherapy [198]. In contrast to the clinical benefit of melanoma patients (see the approved use of ipilimumab, nivolumab and relatlimab, as described above), the clinical response rate of CRC patients was somehow disappointing. Another problem arose when the clinicians realized that an immune checkpoint blockade may induce severe side effects. These immune-related adverse events quite often included colitis (inflammation of the colon) [199]. Nevertheless, clinical research went on, and the use of immune checkpoint inhibitors for the treatment of CRC is under evaluation, as outlined in the following subchapters.

#### 4.1.1. Checkpoint Inhibitor Blockade in dMMR-MSI CRCs

The clinical use of immune checkpoint inhibitors was tested in a huge number of clinical studies, and the anti-tumor efficacy of the treatment regimen very much depends on the different cancer types of the patients. There is no doubt that the most successful anti-tumor story can be told for malignant melanoma [188]. In 2015, a study was published showing that PD-1 blockade was very efficient in patients suffering from mismatch repair-deficient colorectal cancers (dMMR CRC). The authors directly compared the immune-related objective response rate and immune-related progression-free survival rate of dMMR and pMMR CRCs and found that both indices were very much improved in dMMR CRCs but not in pMMR CRCs [5]. In line with this, it was suggested that the microsatellite instable subset of colorectal cancer is particularly responsive to immune checkpoint blockade [100]. This assumption was verified in an expanded 4-year follow-up study, and the data confirmed the long-term benefit of combination therapy using nivolumab plus low-dose ipilimumab for patients with dMMR-MSI CRCs [200]. However, dMMR-MSI CRCs only represent a minority of sporadic CRCs, and it is, therefore, necessary to intensify the research for new therapeutic options to treat metastatic CRC, especially microsatellite-stable CRCs.

#### 4.1.2. Checkpoint Inhibitor Blockade in pMMR-MSS CRCs

As mentioned in Section 1.1, about 85% of sporadic CRCs can be classified as pMMR-MSS. Thus, their mutational load is not very high, and reactivation of immune cells, e.g., cytotoxic T lymphocytes or T helper-1 cells, by immune checkpoint blockade [48] should come into nothing. On the other hand, a recent study demonstrated that the combination of avelumab plus regorafenib mobilizes anti-tumor immunity in patients with microsatellite-stable CCR [201]. Thus, it might be necessary to treat pMMR-MSS CRC patients according to a two-hit model by simultaneously targeting growth-factor-related, tyrosinkinase-dependent signaling pathways and the PD-1/PD-L1 axis of negative immune regulation. In another approach, the combination of nivolumab and urelumab was tested in a preclinical setting in immunodeficient mice. Here, the authors demonstrated that this treatment leads to increased numbers of activated IFN-γ-producing T lymphocytes and decreased numbers of regulatory T lymphocytes. As a result, the therapeutic regimen reduced the tumor growth of transplanted colorectal carcinoma cells [202]. Similarly, in mice transplanted with colorectal cancer cells, the combination of an anti-LAG-3 antibody plus an anti-PD-1 antibody revealed an enhanced control of the tumor growth as compared with the single treatments alone [203]. In a recent proof of concept study, patients with pMMR-MSS metastatic CRC were treated with the chemotherapeutic drug temozolomide, which leads to MMR deficiency and increased tumor mutational burden. After this priming therapy, the patients received checkpoint inhibitor therapy against PD-1 with the antibody pembrolizumab [204]. Taken together, the search for an efficient regimen to treat pMMR-MSS CRC is still ongoing. However, the smart combination of drugs, such as kinase inhibitors, and biologicals, such as monoclonal antibodies against immune checkpoints or growth factor receptors, may be the breakthrough for future treatment algorithms for this devastating disease. This is reflected by the huge number of clinical trials that are listed on ClinicalTrials.gov. Using the term “Metastatic Colorectal Cancer”, a total number of 1997 studies is found (Search of: Metastatic Colorectal Cancer—List Results—ClinicalTrials.gov, accessed on 2 June 2023).

### 4.2. Cellular Immunotherapies

Cancer immunotherapy started approximately 50 years ago with the characterization of the immune system-related messenger molecule interferon (IFN), which was successfully used as an anti-leukemic substance in owl monkeys [205]. In the beginning, immunotherapy thus followed the “magic bullets” concept, which was originally introduced by Paul Ehrlich. Cellular immunotherapy, as a new cell-based strategy to fight cancer, started about two decades after the introduction of IFNs. In a phase I clinical trial, autologous immune-effector cells from patients with metastatic cancer, including CRC, were transfected with the interleukin (IL)-2 gene, and then re-transferred by repeated intravenous infusions. Although the clinical outcome was negligible, this trial can be regarded as an important proof-of-principle study demonstrating that the adoptive transfer of patient-derived effector immune cells, e.g., cytotoxic T cells, is feasible [206]. “Self or Non-self, this is the question” that the immune system has to resolve. In the following years, researchers thus worked on the specificity of the immune effector cells by genetically modifying and analyzing T cells that express chimeric antigen receptors (CARs). These receptors are transmembrane proteins resembling the normal T cell receptor, including one, two or three T cell signaling endodomains leading to antigen-specific activation of the effector cells (first, second and third generation CARs) [207]. After solving the problems of sufficient transfection and in vitro propagation of the genetically engineered T cells, the first clinical studies were performed. In 2014, 30 children with relapsed acute lymphoblastic leukemia (ALL) received CAR T cells targeting CD19. This treatment regimen was very effective with only moderate side effects: complete remission was achieved in 27 patients (90%), whereas severe cytokine-release syndrome developed in 27% of the patients. The cytokine-release syndrome, however, was successfully treated with tocilizumab, an anti-IL-6 receptor antibody [208]. Indeed, B cell malignancies were the most prominent diseases treated with specifically engineered CAR T cells [209]. Besides CD19, which is expressed on hematologic cells, scientists began to search for cell surface antigens that are overexpressed in solid tumors. The cancer-associated Tn glycoform of MUC1, a neoantigen that is present in a variety of cancers, was found to be a good candidate, and anti-Tn-MUC1 CAR T cells demonstrated target-specific cytotoxicity in xenograft models of pancreatic cancer [210]. Likewise, anti-HER2 CAR T cells can persist in patients for 6 weeks without evident toxicities and show significant clinical efficiency against HER2-positive sarcoma [211]. Interestingly, activated CAR T cells should benefit from simultaneous immune checkpoint blockade as the application of the latter leads to a longer activation phase of the immune effector cells. The first preclinical studies indeed showed that the combination of immune checkpoint inhibitors and CAR T cells enhances the anti-tumor efficacy in mouse models of hematologic and solid tumors [212].

#### 4.2.1. Immunotherapies with CAR T Cells in CRC

As mentioned above, CAR T cells have been mainly designed for the treatment of hematological malignancies, i.e., for CD19-positive B cell malignancies. Nevertheless, CAR T cells were immediately discussed to represent a promising tool to fight therapy-refractory CRC, but its application in CRC needed further exploration. In 2017, the results of a phase I trial were published using CAR T cell therapy in a special cohort of CRC patients with metastases, namely in patients with carcino-embryonic antigen (CEA)-positive cancer cells. The safety and efficacy of the utilized CAR T cells targeting the CEA-positive cancer cells was evaluated, and it was shown that CEA CAR T cell therapy in escalating dosage was well tolerated. In addition, some efficacy in terms of tumor shrinkage and/or induction of stable disease was observed [213]. These promising results led to further clinical research and to the initiation of CAR T cell-based clinical trials with emphasis on NKG2D [214], an activating immune-receptor which is able to recognize tumor cells. In this clinical set up, peripheral blood mononuclear cells (PBMC)s are isolated from patients blood. After T cell enrichment, the cells are expanded using IL-2 and anti-CD3 and transfected with a virus carrying the NKG2D-CAR vector. After an additional IL-2 culture, the enriched NKG2D-CAR T cells are re-infused into the patients. According to this or a similar protocol, CRC patients were included in the following clinical trials (for review, see [214]):(i)NCT03018405 (THINK) using CAR T cells.(ii)NCT03692429 (alloSHRINK) using CAR T cells after chemotherapy.(iii)NCT03370198 (LINK) using CAR T cells by hepatic transarterial infusion.(iv)NCT03310008 (SHRINK) using CAR T cells plus FOLFOX (folinic acid, fluorouracil and oxaliplatin).(v)NCT04107142 (CTM-N2D-101) using CAR γδ T cells.

Specifically, the alloSHRINK trial already showed some positive clinical outcomes with 2 partial responses and 9 stable diseases out of 15 patients. The other trials still await in-depth analysis of the clinical efficacy.

In summary, immunotherapies with immune checkpoint inhibitors or CAR T cells in CRC are still limited to mismatch repair deficient and microsatellite instable tumors as the majority of colorectal cancers exclude the tumor-repressing T cell subtypes or attract tumor-promoting immune cells.

#### 4.2.2. Immunotherapies with γδ T Cells in CRC

Antigen-specific recognition of tumor cells (or infectious agents) is classically mediated by a T cell receptor (TCR) consisting of two subunits, the alpha (α)- and the beta (β)-subunit—which are disulfide-linked and associated with a T3 complex. Consequently, T cells expressing an αβ TCR are named αβ T cells (for details, see also Section 1.3.1 and Section 1.3.2). In addition to the ordinary TCR, the product of the TCR gamma (γ) gene was uncovered [215], and the expression of a specific TCR consisting of a γ- and a delta (δ)-chain disulfide-linked to form a heterodimer were described in T cells of the developing thymus [216]. γδ T cells are, therefore, T cells that express a γδ TCR on their surface. Interestingly, the therapeutic potential of γδ T cells was recognized early after their first description in the developing thymus. Malkovska et al. demonstrated significant anti-tumor activity of human γδ T cells in immune-deficient mice, which were inoculated with human Burkitt lymphoma (Daudi) cells. The antilymphoma effect of the γδ T cells was presumably due to groEL (a protein that belongs to the chaperonin family of molecular chaperones) homolog-dependent lysis of the Daudi cells [217]. Similar to the CAR T cells, the originally described antilymphoma effect of the γδ T cells had to be translated in the fight against solid tumors. It took another 10 years before Chen et al. reported on the in vitro and in vivo anti-tumor activity of expanded human tumor-infiltrating γδ T cells against colorectal and ovarian epithelial carcinoma in BALB/c nude mice [218]. In 2010, γδ T cells were brought into the clinics, and it was demonstrated that γδ T cell immunotherapy could be safe and feasible for patients suffering from recurrent non-small-cell lung cancer. However, the clinical outcome was quite disappointing as there were neither complete nor partial responses in any patient, and stable diseases in only three out of ten patients [219]. In another phase I study, it was shown that the efficacy of combination therapy with gemcitabine and autologous γδ T cells in pancreatic cancer patients very much depended on the high quality of the γδ T cell product (>80% γδ T cells) [220]. As γδ T cell can be easily expanded using synthetic antigens, e.g., pyrophosphomonoesters and nitrogen-containing bisphosphonates, and express high levels of PD-1, this special cell type may be useful for combination therapy harnessing the transfer of γδ T cells and immune checkpoint blockade (for review, see [221]). In a recent paper, an overview was provided of small-scale clinical trials with CRC patients employing ex vivo-expanded γδ T cells [144]. Here, the authors summarized that, altogether, only 13 CRC patients have been treated so far with γδ T cells in four different clinical settings. In these studies, no dose-dependent toxicity was found, but the anti-tumor efficacy of the therapy was also negligible.

Taken together, tumor-infiltrating γδ T cells represent a promising tool for T cell-based immunotherapies [222]; however, their usage in clinical settings, especially in the context of CRC, is still experimental, and clinical application awaits approval by the European and non-European authorities.

## 5. Therapeutic Options: Turning “Cold” CRCs into “Hot” CRCs

### 5.1. Possibilities to Increase the Infiltration of T Cells

Compared to “cold tumors”, “hot tumors” are more responsive to immune checkpoint inhibitor (ICI) monotherapy. Thus, promoting the conversion of “cold tumors” to “hot tumors” through interventions can help to reduce resistance to ICI. There are several options to enhance the infiltration of T cells that were verified in preclinical and clinical studies.

Figure 4 summarizes some of the possible options to achieve clinical response discussed in this review.

#### 5.1.1. Oncogenic Pathway Inhibitors

The application of inhibitors that target oncogenic signaling pathways may potentially alleviate the T-cell exclusion commonly observed in tumors. P21-activated protein kinase 4 (PAK4) is known to be highly expressed in “cold tumors” and plays a significant role in the WNT/β-catenin pathway. In a mouse tumor model, the use of KPT-9274, a specific inhibitor of PAK4, or knocking down PAK4 expression, resulted in an increase in cytotoxic T-lymphocyte infiltration within tumors, leading to improved therapeutic outcomes of PD-1 blockade [223]. Nevertheless, the effectiveness of treatments targeting the WNT pathway remains a subject of debate. For instance, certain endogenous inhibitors of the WNT pathway, such as proteins from the Dickkopf (DKK) family, have been found to play a role in promoting tumor immune evasion and have been associated with a poorer prognosis in certain cancers [224]. DKK2, one of these inhibitors, hinders WNT-β-catenin signaling by binding to the cell surface receptors LRP5 and LRP6 of the WNT pathway. Its expression is increased in human CRCs, contributing to tumor progression by suppressing the activation of NK cells and CD8^+^ T cells [225]. These findings challenge the assumption that inhibiting the WNT pathway would enhance immunotherapy. Additionally, recent studies have demonstrated that activating the WNT pathway in endothelial cells promotes T-cell infiltration into tumors and enhances the efficacy of immunotherapies, such as adoptive cell transfer (ACT), suggesting the necessity for further investigation into the feasibility of using WNT inhibitors as immune adjuvants.

In another context, Cyclin D binds to cyclin-dependent kinase 4/6 (CDK4/6), facilitating cell entry into the S-phase through the Retinoblastoma-E2F (RB-E2F) pathway and promoting tumor cell proliferation. Abemaciclib, a CDK4/6 inhibitor, has shown potential in augmenting tumor-infiltrating T lymphocytes and enhancing T-cell activity in CT26 syngeneic mouse tumors. This effect is evidenced by the upregulation of activation markers, including IFN-γ, Granzyme-B, CCL-4 and CCL-5 [226].

#### 5.1.2. Anti-Angiogenic Therapy

One of the hallmarks of tumors is the persistence of angiogenesis, which is regulated by a delicate balance of pro- and anti-angiogenic factors [227]. One crucial factor in this process is vascular endothelial growth factor (VEGF), which plays a significant role in colorectal cancer. The use of bevacizumab, an anti-VEGF antibody, has proven to be effective in inhibiting VEGF and has become an indispensable treatment for metastatic CRC [228,229]. Other anti-angiogenic drugs such as ramucirumab, aflibercept, and regorafenib have also shown promising responses and tolerability in large phase III studies, leading to their approval by the FDA [230,231]. However, the efficacy of anti-angiogenic therapies is somewhat limited, falling short of the high expectations generated by preclinical studies. Clinical trials evaluating sorafenib, sunitinib, vandetanib, and vantalanib as treatments for metastatic colorectal cancer did not yield promising results. Consequently, the clinical development of these drugs for mCRC was halted [232]. Presently, there are ongoing clinical investigations involving novel anti-angiogenic drugs with alternative mechanisms of action that differ from VEGF(R) inhibition [233].

#### 5.1.3. TGF-β Inhibitors

The inhibitory role of TGF-β in immune function has led to the validation of TGF-β inhibition as an effective strategy for promoting T lymphocyte infiltration. TGF-β is associated with a non-inflamed T-cell phenotype, which is characterized by a lack of immune response. Among the various compounds tested, galunisertib, a small molecule that targets the TGF-β receptor 1 (TGFBR1) kinase activity, has been extensively studied [234]. In mouse models of colorectal cancer, galunisertib treatment resulted in increased T-cell infiltration and improved responsiveness to checkpoint therapy. Furthermore, TGF-β hampers the generation of in situ tumor vaccines following radiotherapy. However, treatment with the 1D11 antibody, which blocks systemic TGF-β activity, has been shown to enhance the initiation of T-cell responses to endogenous tumor antigens after subcutaneous tumor irradiation [235].

#### 5.1.4. CXCR4 Inhibitors

The receptor CXCR4 binds to its ligand CXCL-12, which is commonly overexpressed in various types of tumors. The CXCL-12/CXCR4 axis indirectly contributes to the sequestration of cytotoxic T lymphocytes away from the tumor site, leading to reduced infiltration of CTLs. Additionally, this axis facilitates the infiltration of immunosuppressive cells into tumors, promoting an immunosuppressive microenvironment [236]. In a model of pancreatic ductal adenocarcinoma (PDAC), inhibiting the CXCL-12/CXCR4 axis mediated by cancer-associated fibroblasts (CAFs) using the CXCR4 inhibitor AMD3100 resulted in enhanced accumulation of T cells and regression of cancer [237].

### 5.2. Enhancement of Neo-Antigens

Studies have demonstrated that DNA methyltransferase inhibitors (DNMTi) and histone deacetylase inhibitors have the ability to enhance the expression of tumor antigens, components of antigen processing and presenting machinery pathways, and other immune-related genes [238]. These agents can also induce the expression of retroelements, such as endogenous retroviruses (ERVs), which are typically silent but capable of triggering a type I interferon (IFN-α and IFN-β) response [239]. Epigenetic drugs have been found to induce transcription from normally repressed ERV long terminal repeats (LTRs), leading to the production of immunogenic peptides through canonical or novel open reading frames [240]. Moreover, DNMTi and inhibitors of the histone-lysine N-methyltransferase EZH2 have been shown to reverse the epigenetic silencing of Th1-type chemokines in tumor cells. This silencing is negatively associated with the presence of CD8^+^ T cells in tumors and patient outcomes [241]. There is, thus, a strong rationale to combine epigenetic therapy and immunotherapy and many clinical trials are currently ongoing. In colorectal cancer, a study combined the DNMTi drug Azacitidine with the anti-PD-1 antibody Pembrolizumab and the inhibitor of indoleamin 2,3-dioxygenase-1 Epacadostat (NCT02959437). Another study used the DNMTi drug Decitabine together with the EGFR inhibitor Panitumumab (NCT00879385).

### 5.3. Better Priming and Activation of T Cells

#### 5.3.1. Oncolytic Viruses (OVs)

Oncolytic viruses (OVs) have gained recognition as emerging therapeutics due to their potent anticancer activity. Along with inducing selective tumor lysis, OVs can activate both innate and adaptive immune responses, leading to alterations in the tumor microenvironment (TME) [242]. During immunogenic cell death (ICD), three damage-associated molecular patterns (DAMPs) are released: high mobility group box 1 (HMGB1), extracellular adenosine triphosphate (ATP) and cell surface-expressed calreticulin (CRT), which act as adjuvants to enhance dendritic cell (DC) uptake and cross-presentation of tumor antigens to T lymphocytes in draining lymph nodes (DLNs) [243]. OVs also promote the function of DCs by stimulating their production of type I interferons (IFN-α and IFN-β). Immune adjuvants interact with tumor antigens within the tumor and function as personalized in situ vaccines to facilitate T-cell priming [244]. Furthermore, OVs stimulate the production of CXCL-9 and CXCL-10 and upregulate the expression of selectins and integrins, providing crucial signals for T-cell trafficking. Additionally, OVs induce the degradation of the extracellular matrix (ECM), thereby disrupting the physical barrier to T-cell infiltration [244].

#### 5.3.2. Chemotherapy and Radiotherapy

Radiation therapy plays a role in promoting the migration of effector T lymphocytes to the tumor site by inducing the expression and release of chemokines, such as CXCL-10 and CXCL-16, from tumor cells [245]. Fractionated radiotherapy, delivered in individual doses of less than 8–10 Gy, can effectively induce immunogenic cell death (ICD) without increasing hypoxia or immunosuppression, thereby eliciting a de novo anti-tumor response. Preclinical studies have shown that stereotactic body radiotherapy (SBRT) enhances the infiltration of effector T cells into tumors and draining lymph nodes, leading to improved survival rates. Furthermore, in addition to radiation therapy, several chemotherapeutic agents have immune-stimulatory effects by enhancing immunogenicity and increasing T-cell infiltration. Chemotherapy that induces ICD has been demonstrated in various mouse models to convert “cold tumors” into “hot tumors” in response to immune checkpoint inhibitors [246].

#### 5.3.3. Cancer Vaccines

Cancer immunotherapy encompasses cutting-edge strategies for combating cancer, such as monoclonal antibodies (mAb), immune checkpoint inhibitors, adoptive cell transfer (ACT), and cancer vaccines. Notably, sipuleucel-T was the first cancer vaccine approved by the Food and Drug Administration (FDA) in 2010, which has significantly contributed to advancements in the field of cancer vaccines [247]. However, as monotherapy, therapeutic vaccines have not shown significant efficacy in CRC, particularly in advanced stages [248]. A clinical phase 2 study in MMRp colorectal cancer patients using whole-cell cellular immunotherapy as a vaccine showed no objective responses (NCT02981524). This limitation can be attributed to suboptimal antigen selection, adjuvant choice, vaccine platform, and/or improper delivery methods. In a previous study with a vaccine composed of 7 tumor-associated antigens, longer progression-free survival for patients receiving multiple doses of the vaccine compared with patients who received only one dose was observed (NCT03391232).

### 5.4. Enhancement of T Cell Trafficking to the Tumor

Epigenetic modification inhibitors have shown promising results in transforming tumors from an immune “cold” state to an immune “hot” state by various mechanisms. These drugs enhance the expression of multiple chemokines, including CXCL-9, CXCXL-10, and CCL-5, promoting T-cell trafficking to tumors [241]. Epigenetic therapy can also induce ERVs, suppress MYC signaling, and increase the expression of type I IFNs and related chemokines, thereby enhancing tumor immunogenicity. Moreover, epigenetic therapies can restore MHC-I antigen processing and presentation mechanisms and increase the expression of tumor antigens, such as cancer antigens [249]. Among the various non-coding RNAs (ncRNAs), micro RNAs (miRNAs) show the most promise as future biomarkers for CRC. The field of miRNA research is rapidly growing, and their clinical implications are expected to become more prominent within the next decade. Importantly, epigenetic changes are reversible, presenting attractive targets for future cancer treatments. The utility of epigenetic modifiers has been demonstrated in preclinical and phase I/II studies, further highlighting their potential as therapeutic interventions for CRC [250].

In summary, there are several options to enhance T cell infiltration into the tumor, turning “cold” tumors into “hot” tumors, such as using oncolytic viruses, vaccinating against tumor antigens or inhibiting tumor vasculature. Most of these options were very successful in preclinical studies of CRC but often failed in clinical studies as monotherapy or had severe side effects.

## 6. Concluding Remarks

Colorectal cancer is a very heterogeneous type of disease, ranging from highly T cell-infiltrated dMMR, MSI-H tumors to T cell deserted and fibrotic subtypes with a high myeloid score. There are several new therapeutic options, such as immune checkpoint blockade, but most of these therapeutic strategies do not target the tumor cell itself but cells of the tumor microenvironment, primarily infiltrating CD8^+^ cytotoxic T cells, γδ T cells and T helper 1 cells. Cancer cells use several mechanisms to avoid killing by T cells. As cytotoxic T cells are contact-dependent killer cells, one mechanism is to exclude T cells from the tumor. The antigen-specific activation of naïve T cells occurs in secondary lymphoid organs such as tumor-draining lymph nodes and not within the tumor tissue. To infiltrate the tumor, activated T cells have to upregulate integrins and chemokine receptors and cells in the tumor have to produce the corresponding chemokines. Infiltration of an activated T cell into the tumor is a complex process, and many cells within the tumor microenvironment can help to limit the accumulation of the anti-tumorigenic T cell and favor the infiltration of pro-tumorigenic T cells and other immune cells of the tumor microenvironment that protect cancer cells. We discuss several options to enhance the infiltration of anti-tumoral T cells and to turn a “cold” colorectal tumor into a “hot” colorectal tumor (summarized in Figure 4 of this review).

T cells that successfully infiltrate the growing tumor usually have an activated phenotype. In our manuscript, we describe how tumor-infiltrating T cells become anergic, tolerant or immunologically ignorant in tumors that hardly express neo-antigens like most CRCs. In tumors with a high mutational burden, T cells are constantly stimulated by high amounts of neo-antigens over a long period. This leads to a dysfunctional state called exhaustion, to the upregulation of multiple inhibitory checkpoint receptors and finally to terminally exhausted T cells. Terminally exhausted T cells lose the cytotoxic phenotype as well as the ability to proliferate.

Another mechanism tumor cells use to avoid killing by T cells is the downregulation of MHC class I molecules, making the tumor cell “invisible” to conventional T cells with an MHC-I-restricted T cell receptor. There are two subsets of T cells, which do not depend on tumor antigens presented on MHC class I, T cells with a γδ T cell receptor and NKT cells. We discuss the therapeutic options, such as immune checkpoint inhibitor therapy, CAR T cell therapy or immunotherapy with γδ T cells, to overcome dysfunction and restore the killing functions of T cells. In colorectal cancer, emphasizing only one therapeutic option does not seem to be successful. The limited effects of checkpoint inhibition in microsatellite-stable, MMR-proficient CRCs show this dilemma. Tumors that are nearly devoid of T cells with a cytotoxic phenotype are not good targets for checkpoint inhibition alone. Additionally, in solid tumors, even the application of CAR T cells as a monotherapy showed only reduced effects. However, most clinical studies combine several therapeutic modalities to achieve clinical responses even in tumors that are hard to treat, e.g., colorectal cancer.

## Figures and Tables

**Figure 2 ijms-24-11673-f002:**
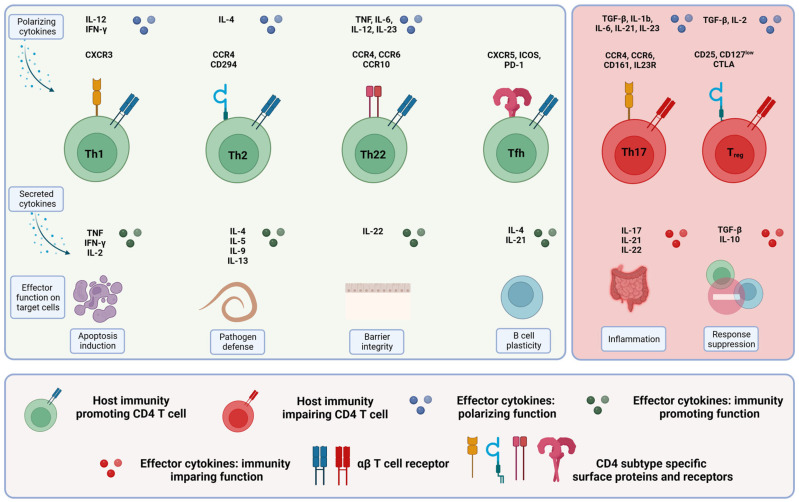
CD4^+^ T cell subsets in colorectal cancer. The different immunity-promoting CD4^+^ T cell subsets in colorectal cancer are shown in green (left), and the immunity-impairing CD4^+^ T cell subsets are shown in red (right). The morphology, including the specific surface proteins and receptors (upper row), the secreted cytokines (middle row) and the respective effector functions (lower row) of the different CD4^+^ T cell subsets, are given. For more details, see text.

**Figure 4 ijms-24-11673-f004:**
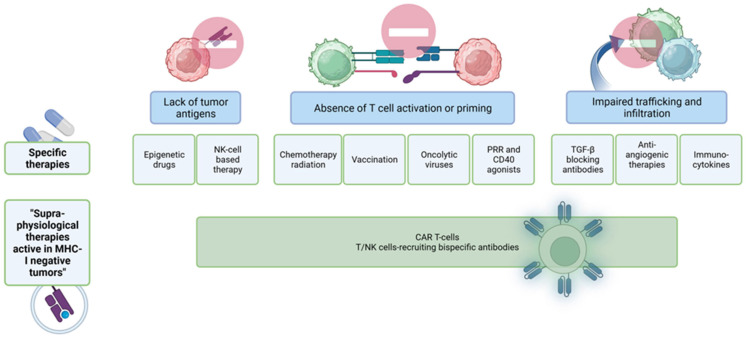
Options to turn a “cold tumor” into a “hot tumor”. There are several strategies to overcome the lack of tumor antigens, the absence of T cell activation and priming or the absence of tumor-infiltrating T cells within colorectal tumors. Some therapies target the epigenomic landscape of CRC, others use antibodies for different pathways involved in cancer progression or try to normalize the tumor vasculature by anti-angiogenic molecules. Another option is the application of bispecific antibodies that recognize T cells or NK cells and tumor-specific antigens on the surface of tumor cells.

**Table 1 ijms-24-11673-t001:** Characterization of the main CD4^+^ T cell subsets in colorectal cancer.

Name	Surface Marker	Transcription Factor	Polarizing Cytokines	Function	Secreted Cytokines	References
Th1	CXCR3^+^, CCR6^−^	T-BET	IL-12, IFN-γ	Destruction of infected cells by inducing apoptosis. Th1 cells promote destruction and induce senescence of cancer cells. They also display anti-angiogenic properties.	IFN-γ, TNF-α, IL-2	[46,47,48]
Th2	CXCR3^−^, CCR4^+^, CCR6^−^, CD294^+^	GATA	IL-4	Response to extracellular pathogens. Polarization of macrophages toward a tumor-promoting M2 phenotype. Th2 cells promote the proliferation of tumor cells.	IL-4, IL-5, IL-9, IL-13	[49]
Th17	CXCR3^−^, CCR4^+^, CCR6^+^, CD161^+^, IL23R^+^	IRF4^+^, ROR-γt^+^	TGF-β, IL-6, IL-1β, IL-21, IL-23	Response to extracellular pathogens by recruiting neutrophils and macrophages to the site of inflammation. Promotion of cancer stemness and chemo-resistance.	IL-17, IL-21, IL-22	[46,50]
Th22	CCR10^+^, CCr4^+^, CCR6^+^	AHR^+^, FOXO4^+^	IL-6, TNF-α, IL-12, IL-23	Regulator of epithelial barrier integrity.	IL-22	[51]
Tregs	CD127^low^, CD25^+^, CTLA4^+^	FOXP3^+^	TGF-β, IL-2	Maintain tolerance to self-antigens; prevent the induction of autoantibodies. Suppression of effector T cell-mediated immune responses in colorectal cancer.	IL-10, TGF-β	[46,52]
Tfh	CXCR5^+^, ICOS^+^, PD-1^+^	BCL6^+^		Orchestration of germinal center B cell responses; required for antibody class switching.	IL-21, IL-4	[53]

## Data Availability

Not applicable.

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
