# Peer review of "T Cells in Colorectal Cancer: Unravelling the Function of Different T Cell Subsets in the Tumor Microenvironment"

_ijms, 2023, doi:10.3390/ijms241411673_

Round 1

Reviewer 1 Report

The review entittled "T cells in Colorectal Cancer: Unravelling the function of different T cell subsets in the tumor microenvironment" by Zheng et al is a rich in information and valuable piece of work that summarizes the state of the art in the field.

My recommendations:

1. There are several acronyms used in the review, a list of abbreviations would be helpful.

2. It is compact and at times difficult to follow for the ones who are not 100% in the domain. To make the review more accessible to a greater public:

      i) provide 'take home messages' for each section

     ii) provide a short first section along with a Figure demonstrating the basics of immunology and molecular biology related to this topic.

3) For any mAb link to IMGT/mAb-DB and the relevant info included in it. For instance, line 628, ipilimumab, link to https://www.imgt.org/mAb-DB/mAbcard?AbId=180  and the video: https://www.imgt.org/video/IMGT_MOA_2.mp4

4) Provide a graphical abstract/ video summarizing and illustrating the main points.

Author Response

Reviewer 1:

The review entittled "T cells in Colorectal Cancer: Unravelling the function of different T cell subsets in the tumor microenvironment" by Zheng et al is a rich in information and valuable piece of work that summarizes the state of the art in the field.

Thank you very much for this positive feedback.

My recommendations:

  1. There are several acronyms used in the review, a list of abbreviations would be helpful.

This is a very sensible suggestion as readers that are not so familiar with tumor immunology and especially with colorectal cancer immunology and therapeutic strategies get overwhelmed by the mere quantity of abbreviations. We added a list of abbreviations at the end of the revised manuscript (Lines 974-1045).

  1. It is compact and at times difficult to follow for the ones who are not 100% in the domain. To make the review more accessible to a greater public:
  2. i) provide 'take home messages' for each section

Done as suggested. Several “take home messages” are now provided as short paragraphs at the end of the respective chapter of the revised manuscript (marked in red).

  1. ii) provide a short first section along with a Figure demonstrating the basics of immunology and molecular biology related to this topic.

As suggested by the reviewer, we introduce the topic of immunology at the very beginning of our text and refer to a widely used textbook of immunobiology and a well-known review on Cancer Immunology. Both references contain a whole bunch of very informative figures. At the beginning of the revised version of our manuscript, we now write:

The immune system is the body´s defence machinery against harmful influences originating either from an infection or from malignant transformation of the body´s own cells. The vertebrate immune system consists of a variety of innate and adaptive effector cells, e. g. granulocytes, macrophages, dendritic cells, T and B lymphocytes etc., and specialized molecules, e. g., T cell receptors, specific antibodies, Toll-like receptors, interleukins etc. (for an overview in immunology and cancer immunology, the interested reader is referred to Kenneth Murphy and Casey Weaver (2017) Janeway´s Immunobiology, 9th edition, New York, NY, Garland Science, Taylor and Francis; Karin E. de Visser and Johanna A. Joyce (2023) The evolving tumor microenvironment: From cancer initiation to metastatic outgrowth, Cancer Cell 41: 374-403; Olivera J. Finn (2008) Cancer Immunology, N. Engl. J. Med. 358: 2704-15). In the current review, we will focus on the role of T cells in the context of CRC” (lines 46-51).

3) For any mAb link to IMGT/mAb-DB and the relevant info included in it. For instance, line 628, ipilimumab, link to https://www.imgt.org/mAb-DB/mAbcard?AbId=180  and the video: https://www.imgt.org/video/IMGT_MOA_2.mp4

As suggested, we inserted after the list of abbreviations a link to IMGT/mAb-DB for any mAb mentioned in the text (lines 1048-1059).

4) Provide a graphical abstract/ video summarizing and illustrating the main points.

We thank the reviewer for this suggestion but think that a graphical abstract is very useful for a data manuscript that analysis some aspects of T cells in colorectal cancer but is much too complex for a review, dealing with all aspects of T cells in CRC. In our opinion, a graphical abstract cannot reflect the complexity of the topic of our review. We have therefore put much emphasis on the written abstract where we describe all aspects of T cell-mediated effects in CRC including the therapeutical potential using immune checkpoint inhibitors or adoptive cell transfer. In addition, Figures 1, 2, 3 and 4 already represent graphical abstracts of the different chapters of the review. In our opinion, a graphical abstract that reflects every aspect of T cells in colorectal cancer including immunotherapeutic interventions and therapeutic options would induce more confusion than clarification.

Reviewer 2 Report

This is a timely review of CRC and T cells activity when recruited to the tumor microenvironment (TME). Literature revised is recent and classical which provides a very good picture of how CRC immune research has been growing up, and most importantly, how this important type of cancer is different from other solid tumors and how the same type of intervention such as ICB does not work on CRC. This definitely shows the necessity for more specific research on CRC.

The very nice explanation of how CRC is classified is worthy. 

Minor points:

1. There are a few typo mistakes and a lack of specificity on TNF and IFN, please clarify if you refer to TNF-alpha and IFN-gamma along the text.

2. Figure 3 is not necessary, it looks like a text book figure of basic immunology.

3. CD4 CTLs are still non-completely accepted in the immunology field, so I suggest removing this section or giving emphasis that is not a population fully recognized.

Just type mistakes such as sometimes you write IL-... iL-... or IL

Author Response

Reviewer 2:

This is a timely review of CRC and T cells activity when recruited to the tumor microenvironment (TME). Literature revised is recent and classical which provides a very good picture of how CRC immune research has been growing up, and most importantly, how this important type of cancer is different from other solid tumors and how the same type of intervention such as ICB does not work on CRC. This definitely shows the necessity for more specific research on CRC.

The very nice explanation of how CRC is classified is worthy. 

Thank you very much for this positive feedback.

Minor points:

  1. There are a few typo mistakes and a lack of specificity on TNF and IFN, please clarify if you refer to TNF-alpha and IFN-gamma along the text.

In the revised manuscript, we corrected the typos throughout the text. The use of TNF and IFN is now clarified.  

  1. Figure 3 is not necessary, it looks like a text book figure of basic immunology.

We agree with the reviewer that Figure 3 mainly depicts basic knowledge about the cytotoxic branch of cellular immunity. For reasons of consistency (see Figure 2 that shows the basic knowledge about CD4-positive T cells), we decided to keep this Figure in the revised version of the manuscript.

  1. CD4 CTLs are still non-completely accepted in the immunology field, so I suggest removing this section or giving emphasis that is not a population fully recognized.

We agree with the reviewer that CD4+ CTLs represent a cell population that is not fully accepted and have been mainly described in the context of malignant melanoma. As this cell type obviously does not play an important role in CRC, we removed the section about CD4+ CTLs from the revised version of the manuscript.  

Round 2

Reviewer 1 Report

IMGT/mAb-DB should be cited:

Manso T., Kushwaha A., Abdollahi N., Duroux P., Giudicelli V. and Kossida S. Mechanisms of action of monoclonal antibodies in oncology integrated in IMGT/mAb-DB. Front Immunol., 14 (2023). DOI 10.3389/fimmu.2023.1129323